# Optimizing Pediatric Intermediate Care: Clinical Predictors of Deterioration and Length of Stay in a Tertiary Setting

**DOI:** 10.3390/jcm14186398

**Published:** 2025-09-10

**Authors:** Giacomo Brisca, Carlotta Pepino, Marcello Mariani, Giacomo Tardini, Marta Romanengo, Emanuele Giacheri, Marisa Mallamaci, Isabella Buffoni, Valentina Carrato, Marina Francesca Strati, Stefania Santaniello, Rossana Taravella, Laura Puzone, Lisa Rossoni, Michela Di Filippo, Daniela Pirlo, Andrea Moscatelli

**Affiliations:** 1Neonatal and Pediatric Intensive Care, Intermediate Care Unit, Emergency Department, IRCCS Istituto Giannina Gaslini, 16147 Genoa, Italy; giacomotardini@gaslini.org (G.T.); martaromanengo@gaslini.org (M.R.); emanuelegiacheri@gaslini.org (E.G.); marisamallamaci@gaslini.org (M.M.); isabellabuffoni@gaslini.org (I.B.); valentinacarrato@gaslini.org (V.C.); marinafrancescastrati@gaslini.org (M.F.S.); stefaniasantaniello@gaslini.org (S.S.); danielapirlo@gaslini.org (D.P.); andreamoscatelli@gaslini.org (A.M.); 2Emergency Room and Pediatric Emergency Medicine, Emergency Department, IRCCS Istituto Giannina Gaslini, 16147 Genoa, Italy; carlottapepino@gaslini.org; 3Pediatric Infectious Diseases Unit, IRCCS Istituto Giannina Gaslini, 16147 Genoa, Italy; marcellomariani@gaslini.org; 4Department of Neurosciences, Rehabilitation, Ophthalmology, Genetics and Maternal and Child Health (DINOGMI), University of Genova, 16147 Genoa, Italy; ross.taravella@gmail.com (R.T.); lauretta.puzone2@gmail.com (L.P.); lrossoni.22@gmail.com (L.R.); michela.difilippo1@gmail.com (M.D.F.)

**Keywords:** risk management, clinical deterioration, adverse outcome, length of stay, unplanned transfer

## Abstract

**Background/Objective:** Pediatric Intermediate Care Units (PIMCUs) provide enhanced monitoring and support for children who require more care than standard wards but do not meet full Pediatric Intensive Care Unit (PICU) criteria. Despite their growing role, evidence on how to stratify risk and predict clinical trajectories within this specific population remains scarce. This study aimed to identify admission factors associated with (1) early unplanned transfer to the PICU within 48 h and (2) prolonged length of stay (LOS) in the PIMCU of a tertiary Italian pediatric hospital. **Methods:** We conducted a retrospective observational study including 893 children admitted to the PIMCU at IRCCS Gaslini Children’s Hospital (Genoa, Italy) between January 2022 and June 2023. Demographic, clinical, laboratory, and outcome data were collected. Multivariable logistic regression and negative binomial models were used to assess predictors of early PICU transfer and prolonged LOS, respectively. **Results:** Early PICU transfer occurred in 2.8% of cases. Tachypnea (OR = 2.80; *p* = 0.018) and nasogastric tube (OR = 3.72; *p* = 0.014) at admission were independently associated with PICU transfer within 48 h. Prolonged LOS was significantly associated with the need for respiratory support and the presence of medical devices, including nasogastric tubes, central venous lines, and thoracic/abdominal drains. **Conclusions:** Specific clinical markers and device use at admission can help identify patients at higher risk of deterioration or extended PIMCU stay, supporting more accurate triage, early intervention, and resource optimization in pediatric intermediate care settings.

## 1. Introduction

Pediatric Intermediate Care Units (PIMCUs) play a crucial role in managing children who require closer monitoring than in general pediatric wards but do not yet require full Pediatric Intensive Care Unit (PICU) support [1,2]. Their relevance has increased with the growing complexity of pediatric illnesses and technological advancements that enable better outcomes for critically ill children while easing pressure on PICU resources.

The definition of PIMCU remains heterogeneous. Also referred to as high-dependency, progressive, or step-up units, PIMCUs offer intensive observation and treatment for children at risk of physiological deterioration. They can also function as step-down units for patients recovering from critical illness or surgery [3].

By bridging the gap between general wards and intensive care, PIMCUs offer higher nurse–patient ratios, specialized staff, and support for interventions such as non-invasive ventilation [4]. Patient safety is a key target for PIMCU activity. Ensuring patient safety involves the timely recognition of deterioration, targeted interventions, appropriate admission criteria, and the efficient allocation of resources.

The American Academy of Pediatrics first published admission and discharge criteria for children requiring admission to PIMCU in 2004 [1], with a recent update in 2022 [3].

Most studies that have explored the reasons for clinical deterioration and the clinical profile of pediatric inpatients have primarily focused on children admitted to general wards or the Emergency Department (ED) [5,6,7,8,9]. These studies highlighted that among children subsequently admitted to intensive care, those transferred from general wards had higher crude mortality rates than those admitted directly from the ED. In response, many pediatric hospitals have implemented Medical Emergency Teams (METs) to facilitate early recognition and treatment of deteriorating children [10,11,12].

An additional, yet underexplored, strategy is the development of Pediatric Intermediate Care Units (PIMCUs), which enable closer monitoring and surveillance, potentially allowing for earlier detection of clinical decline and reducing the need for unplanned PICU transfers.

Despite their growing role, research on the functioning of PIMCUs and the identification of specific predictive factors for clinical deterioration or prolonged hospitalization remains limited [13,14,15]. Understanding which characteristics at admission are associated with adverse outcomes or longer stays is critical both for clinical decision-making and for optimal resource allocation [16].

The present study aimed to evaluate the characteristics of patients admitted to an independent PIMCU in a large tertiary Italian pediatric hospital. Specifically, we sought to identify (1) which initial factors were associated with early unplanned transfer to the PICU within the first 48 h, and (2) which variables at admission correlated with a longer length of stay in PIMCU. These insights may support improved triage decisions, tailored monitoring strategies, and more efficient use of intermediate care resources.

## 2. Materials and Methods

We conducted a retrospective observational single-center cohort study, including all children admitted to the PIMCU of IRCCS Gaslini Children’s Hospital (Genoa, Italy) from 1 January 2022 to 30 June 2023.

For each patient, demographic data (age, sex, origin), clinical characteristics (pre-existing condition, the main reason for admission, vital parameters adjusted for age [blood pressure, breathing rate, heart rate, SatO2, body temperature], respiratory support, medical device, bedside pediatric early warning score [BPEWS]), and blood tests on admission, and 28-day mortality after PIMCU discharge were collected.

Vital parameters were assessed based on the Pediatric Advanced Life Support (PALS) algorithm [17]. Medical devices were included according to the national classification of medical devices (CND) [18]. The BPEWS is a warning score system largely validated as a tool for early recognition of pediatric patient deterioration [19].

The PIMCU was introduced in 2020 as a standalone unit adjacent to a quaternary-level PICU, offering critical care transport and extracorporeal membrane oxygenation retrieval capabilities. Further details on staff, equipment, and organization have already been published [20].

PIMCU receives acute patients from the ED, patients from general pediatric wards or other regional pediatric hospitals requiring care intensification, post-surgical patients from the operating rooms, and children from the PICU as the initial step-down phase of care. When clinical stability is achieved, children are transferred to general wards or may be directly discharged home in case of rapid clinical improvement. Conversely, children who develop clinical deterioration are admitted to the PICU. The admission and discharge criteria for both PIMCU and PICU were developed with input from multiple stakeholders and reflect the availability of resources, clinical experience, and guidance from the American Academy of Pediatrics document on the structuring of PIMCUs [3]. (Appendix A [21,22,23]). The category “Surgical/orthopaedical” included both patients who underwent surgery and those with a surgical or orthopedic condition who did not require an operation.

Descriptive analysis was performed, reporting median and interquartile range (IQR) for all continuous variables due to their non-normal distribution, and absolute and relative frequencies for categorical variables. To identify independent factors associated with IMCU LOS in days, a Generalized Linear Model (GLM) with a Negative Binomial distribution and a log link function was employed, accounting for the count nature and right-skewed distribution of LOS. This analysis was conducted exclusively on the 345 patients who were directly discharged home from the PIMCU, excluding all patients who were transferred to a higher- or lower-intensity unit. To assess risk factors for PICU transfer within 48 h of IMCU admission, initial comparisons between patient groups were conducted using the Mann–Whitney U test for continuous variables and chi-square/Fisher’s exact test for categorical variables. The 48-h cutoff for early transfer was chosen to capture acute and rapid clinical deteriorations, reflecting a critical period of instability commonly recognized in clinical literature on pediatric outcomes [24]. Variables showing a *p*-value < 0.05 in univariable analysis were subsequently included in a multivariable binary logistic regression model. This selection approach was chosen to build a parsimonious model, though we acknowledge its potential limitations in excluding important confounding factors. For both the GLM and logistic regression models, collinearity among predictors was assessed using the Variance Inflation Factor (VIF), with a cutoff of VIF > 5 indicating problematic collinearity. In our multivariable model for PIMCU length of stay, various blood test parameters (e.g., C-reactive protein, procalcitonin, white blood cells, etc.) were initially considered for inclusion. However, these variables were ultimately excluded from the final multivariable model, as their addition negatively impacted the model’s overall fit and explanatory power, resulting in a reduction in the pseudo R-squared. Furthermore, from a clinical standpoint, their independent contribution to predicting length of stay was considered less direct compared to objective physiological or device-related markers present at admission. All statistical analyses were conducted using Jamovi (version 2.6.23) with the GAMLj3 add-on.

## 3. Results

### 3.1. Patient Characteristics and Flow

The flowchart in Figure 1 illustrates the patient selection process and the statistical analyses performed.

Overall, 928 patients were managed in the PIMCU during the study period. We excluded 35 children who had a planned admission directly from their homes, as these were complex patients admitted in anticipation of procedures or surgical interventions and therefore had a markedly higher baseline risk of requiring subsequent admission to intensive care. This cohort of 35 excluded patients had a median age of 14 years (IQR 6–128 months) and included patients with surgical/oncological diseases (n = 12), neurological conditions (n = 10), and other complex pre-existing conditions (n = 13). The remaining 893 patients were included in the analysis, with a median age of 4.0 years (IQR 6–128 months) and a slight predominance of males (55% vs. 45%).

Table 1 summarizes demographic, clinical, laboratory, and outcome details.

Most children came from the Emergency Department (66.7%), followed by patients from general pediatric wards or local hospitals (21.3%) and those transferred from the PICU (11.1%) (Figure 2).

More than half of the primary reasons for PIMCU admission were respiratory (26.7%), neurologic (19.6%), and infectious (13.8%) conditions.

A significant proportion of the PIMCU population (67.5%) had a pre-existing condition, with the most frequent being neurological (17%) and hematological (12.9%) diseases. At least one medical device was present on admission in 31.7% of patients, with central venous catheters being the most common (16.7%).

Overall, children spent a median of 4 days (IQR 2–7) in the PIMCU. Fifty-three (5.9%) patients required PICU admission for further clinical deterioration, 25 (2.8%) of them within the first 48 h. Among the others, 55.9% were moved to a lower-intensity care unit after clinical improvement, and 38.8% were directly discharged home (Figure 2). One patient died during the PIMCU stay in a palliative care program, while 11 (1.2%) died within 28 days of IMCU discharge.

### 3.2. Analysis of Risk Factors for PICU Admission

No significant differences were observed between children who required early transfer to the PICU and those who did not, in terms of age, sex, origin, or reason for admission.

In the multivariate analysis (Table 1) we found that tachypnea adjusted for age (OR = 2.80; 95% CI: 1.19–6.56; *p* = 0.018) and nasogastric tube on admission (OR = 3.72; 95% CI: 1.30–10.57; *p* = 0.014) were independently associated with increased risk of early ICU transfer. The overall predictive model for early PICU transfer demonstrated acceptable discrimination, with an Area Under the Receiver Operating Characteristic (AUC-ROC) curve of 0.678 (95% CI: 0.58–0.78).

No significant associations were found with the BPEWS and lab tests at admission.

### 3.3. Factors Influencing PIMCU Length of Stay

The GLM showed that the need for respiratory support (IRR = 1.36; 95% CI: 1.02–1.81; *p* = 0.035) and the presence of a medical device on admission were strongly associated with prolonged PIMCU hospitalization. Among medical devices, nasogastric tube (IRR = 1.63 (95% CI: 1.15–2.30; *p* = 0.005), central venous line (IRR = 1.73 (95% CI: 1.31–2.27; *p* < 0.001), and thoracic or abdominal drainage (IRR = 3.17 (95% CI: 1.81–5.56; *p* < 0.001) were all independently associated with longer PIMCU stays (Table 2).

## 4. Discussion

Our study analyzed the activity of an independent Pediatric Intermediate Care Unit (PIMCU) in a tertiary Italian pediatric hospital, focusing on identifying admission factors predictive of two critical outcomes: early unplanned transfer to the Pediatric Intensive Care Unit (PICU) and prolonged length of stay (LOS) in the PIMCU. Understanding these aspects is essential for optimizing patient safety, quality of care, and resource allocation.

Unplanned PICU transfers are a sensitive marker of clinical deterioration and can reflect the adequacy of triage decisions and the quality and intensity of care in PIMCUs [15]. Most literature addressing the risk of clinical deterioration has focused on general wards or emergency settings [9,25,26], with less attention paid to the PIMCU population, whose characteristics differ significantly. Our findings provide important insight into this understudied group.

A significant proportion of patients admitted to PIMCU had pre-existing conditions and medical devices, both of which were independently associated with increased risk of PICU transfer and longer PIMCU stays. These findings underscore the PIMCU’s role in managing medically complex children who, while not in critical condition, require a higher level of surveillance and tailored care plans. However, such patients remain at elevated risk of deterioration due to their fragile clinical status [27,28,29].

Among the most relevant predictors of adverse outcomes, tachypnea at admission was independently associated with early PICU transfer. Tachypnea is often one of the earliest and most sensitive indicators of respiratory or systemic distress in children, frequently preceding other signs of clinical deterioration such as hypoxemia, altered mental status, or hemodynamic instability.

In the pediatric population, where compensatory mechanisms are initially effective, an elevated respiratory rate may reflect an early stage of decompensation that is not yet clinically overt. Our observations underline the prognostic value of respiratory rate as a dynamic clinical parameter and support its integration into early warning systems and triage protocols, also within intermediate care settings. Furthermore, this finding may suggest a need for better training of medical and nursing staff on the therapeutic management and monitoring of patients with respiratory impairment and a different allocation of resources.

In addition, the need for respiratory support at admission was a strong predictor of prolonged PIMCU stay. This likely reflects both the underlying severity of the patient’s condition and the resource-intensive nature of managing respiratory insufficiency. Patients requiring respiratory support, especially high-flow nasal cannula or non-invasive ventilation often require extended monitoring, specialized nursing care, and a longer stabilization period before being eligible for step-down or discharge.

These findings emphasize the importance of early identification of respiratory compromise and appropriate allocation of resources, including skilled personnel and equipment, to support these patients effectively within the PIMCU.

The presence of a NGT on admission also emerged as a significant predictor of both adverse outcomes. This observation may reflect the NGT’s role as a proxy indicator of both underlying clinical severity and impaired oral feeding capacity, particularly in patients with respiratory distress, neurological compromise, or hemodynamic instability. The placement of an NGT often indicates the need for nutritional support in patients who are unable to maintain adequate oral intake due to altered mental status, respiratory fatigue, or the risk of aspiration. These conditions are frequently markers of more complex clinical trajectories and may predispose patients to rapid deterioration requiring escalation of care. Furthermore, the presence of an NGT may correlate with other signs of fragility, such as increased work of breathing or persistent vomiting, which are not always fully captured by standard clinical scores but can significantly impact the patient’s stability. Accordingly, the enteral tube has been previously recognized as part of a 7-item score associated with the clinical deterioration of hospitalized children [30].

Interestingly, our analysis did not identify age or laboratory values at admission as independent predictors of adverse outcomes. Younger age is largely known as a critical factor associated with a higher risk of clinical deterioration, according to a combination of several factors (physiological immaturity, higher vulnerability to infections, potentially delayed recognition of serious conditions, etc.) [31,32,33]. Although children < 1 year of age accounted for a relevant part of our PIMCU population (31%), they were not significantly associated with poor outcomes. This emphasizes the role of the PIMCU as an appropriate setting for the care of young children by a healthcare team trained in the interpretation of age-related vital signs and the recognition of early signs of decompensation.

Similarly, while laboratory markers are often used to assess disease severity, they may not capture early physiological changes that precede overt clinical decompensation, highlighting the need for a holistic and dynamic clinical assessment.

One notable finding was that the Bedside Pediatric Early Warning Score (BPEWS) did not emerge as an independent predictor of early PICU transfer. This is an important finding, as such scoring systems are widely used in general wards to identify early deterioration. However, patients in a PIMCU setting, by definition, present with more severe and complex conditions from the start, often with an already elevated baseline score. This higher initial clinical complexity may reduce the discriminatory power of a score like BPEWS, which is more effective at detecting acute changes from a relatively stable baseline. Our findings support a multifactorial approach to risk prediction that integrates vital signs, device presence, and clinical context rather than relying solely on static laboratory values. Implementing structured early warning systems and training teams to interpret early signs of deterioration could enhance the PIMCU’s role in preventing unplanned ICU transfers.

The median PIMCU stay of 4 days aligns with literature values [13,14,34], confirming its function as a setting for intensive yet time-limited care. Our low rate of PICU transfer (5.3%) suggests that patients are appropriately triaged to PIMCU and managed safely, with escalation of care occurring when clinically justified. This is relevant when we examine the benefits of introducing a PIMCU to the PICU’s operations. A safe and well-functioning PIMCU may reduce ICU admissions and overcrowding, favoring a better utilization of critical beds [35]. Furthermore, PIMCU also played a significant role as a step-down unit offering a crucial transition space in the initial phase of care reduction, potentially decreasing ICU stays, and healthcare costs, and enhancing patient comfort.

Lastly, we observed a notable subgroup of patients who were transferred from the PICU and subsequently re-admitted to intensive care (7%). Although not statistically significant, this trend highlights the importance of cautious patient selection and close monitoring during step-down phases. Coordinated communication between PICU and PIMCU teams is essential to ensure safe transitions.

Our study has several limitations, including its retrospective design and heterogeneous patient population. An important limitation is the small number of events (n = 25) for the primary outcome of early PICU transfer, which resulted in a substantial class imbalance compared to the control group (n = 868). This imbalance may affect the stability of the logistic regression model’s findings and limit their generalizability. We did not use techniques such as propensity score matching (PSM). We emphasize that our primary objective was not to build a robust predictive model, but rather to identify factors associated with rapid clinical deterioration within our specific patient cohort. Furthermore, we did not assess the specific treatments required post-transfer to the PICU, nor the reasons for deterioration in detail. The novelty of the PIMCU at our institution may also have influenced our results, as practices and expertise continue to evolve.

We conducted a limited comparison with existing literature, considering that research focusing on pediatric IMCU is minimal. In Italy, this may be linked to the lack of specific regulatory legislation, resulting in significant variability in terms of setting, equipment, and staffing of PIMCUs [36]. Following this, the single-center nature of the study may also limit the generalizability of our findings, because of site-specific practices and policies, including the availability of staff and equipment, and the hospital organization.

Nevertheless, this study contributes to the limited but growing body of knowledge on pediatric intermediate care and highlights practical factors that can improve triage, care delivery, and patient outcomes.

## 5. Conclusions

We identified a set of admission characteristics—including tachypnea, need for respiratory support, and presence of medical devices—that are associated with early unplanned PICU transfer and prolonged PIMCU stay. These findings reinforce the role of clinical observation and early recognition of physiological compromise in guiding patient management. A well-functioning PIMCU contributes to improved outcomes, reduced PICU crowding, and more efficient use of healthcare resources. Further multicenter studies are needed to validate these predictors and develop standardized approaches to pediatric intermediate care.

## Figures and Tables

**Figure 1 jcm-14-06398-f001:**
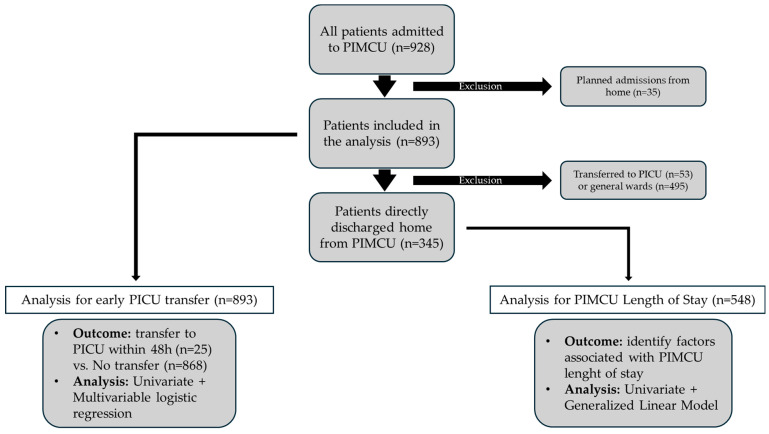
Study design and statistical analysis flowchart. PIMCU, Pediatric Intermediate Care Unit; PICU, Pediatric Intensive Care Unit.

**Figure 2 jcm-14-06398-f002:**
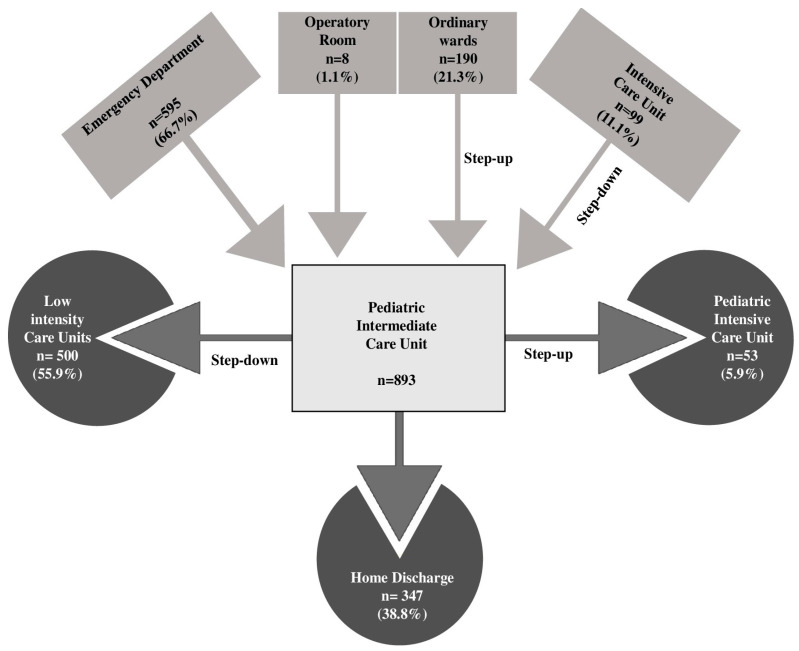
Hospital flow of patients managed in the Pediatric Intermediate Care Unit.

**Table 1 jcm-14-06398-t001:** Demographic, clinical, laboratory, and outcome data of children managed in the PIMCU are presented. Comparisons between those requiring early unplanned transfer to the PICU and those who did not are provided, along with the corresponding results of univariate and multivariable analyses. PIMCU, pediatric intermediate care unit; PICU, pediatric intensive care unit; IQR, interquartile range; CI, confidence interval; OR, odds ratio; HFNC, high-flow nasal cannula; NIV, non-invasive ventilation; VP, ventricular peritoneal; VA, ventricular-atrial drainage; EVD, external ventricular drainage; CRP, C-reactive protein; PCT, procalcitonin; WBC, white blood cells; Hb, hemoglobin. Statistically significant values are expressed in bold.

	Total		PICU Transfer Within 48 h		No PICU Transfer Within 48 h		*p* Univariate	OR Univariate (95%CI)	*p* Multivariate	OR Multivariate (95%CI)
	N	%	N	%	N	%				
**Total patients**	893	100	25	2.8	868	97.2				
**Sex**							0.617			
*Female*	401	45	10	40	391	45				
*Male*	492	55	15	60	477	55				
**Age (months) median IQR**	48	6–128	72	12–185	48	6–27	0.285			
**Department of Origin**							0.149			
*Lower intensity*	190	21	8	32	182	21				
*Emergency department*	596	67	14	56	582	67				
*Operatory room*	8	1	1	4	7	1				
*Pediatric Intensive Care Unit*	99	11	2	8	97	11				
**Reason for admission**							0.467			
*Respiratory*	239	27	10	40	229	26				
*Cardiac*	52	6	0		52	6				
*Infectious*	123	14	1	4	122	14				
*Neurological/psychiatric*	175	20	5	20	170	20				
*Gastroenterological*	45	5	0		45	5				
*Surgical/orthopaedical*	71	8	5	20	66	8				
*Inflammatory disease*	19	2	0		19	2				
*Nephrological/Urological*	11	1	0		11	1				
*Endocrine*	14	2	0		14	2				
*Haemato-oncological*	94	11	3	12	91	11				
*Other*	50	5	1	4	49	5				
**Underlying disease**										
*Respiratory*	51	6	2	8	49	6	0.649			
*Cardiac*	67	8	4	16	63	7	0.111			
*Infectious*	14	2	0		14	2	1.000			
*Neurological/psychiatric*	153	17	7	28	146	17	0.144			
*Gastroenterological*	29	3	1	4	28	3	0.567			
*Surgical/orthopaedical*	37	4	2	8	35	4	0.278			
*Inflammatory disease*	15	2	0		15	2	1.000			
*Nephrological/Urological*	27	3	1	4	26	3	0.541			
*Endocrine*	20	2	0		20	2	1.000			
*Haemato-oncological*	115	13	3	12	112	13	1.000			
*Metabolic disease*	35	4	4	16	31	4	**0.014**	5 (2–16)	0.052	3.29 (0.99–10.94)
*Prematurity*	48	5	1	4	47	5	1.000			
*Other*	84	9	2	8	82	9	1.000			
**Temperature at admission**							0.478			
<37 °C	644	72	18	72	626	72				
<38.5 °C	196	22	7	28	189	22				
>38.5 °C	53	6	0		53	6				
**Hypotension (adjusted for age)**	34	4	2	8	32	4	0.243			
**Tachycardia (adjusted for age)**	138	15	5	20	133	15	0.571			
**Bradycardia (adjusted for age)**	19	2	0		19	2	1.000			
**Tachypnea (adjusted for age)**	214	24	12	48	202	23	**0.008**	**3 (1.4–6.8)**	**0.018**	**2.8 (1.2–6.6)**
**Bradypnea (adjusted for age)**	148	17	2	8	146	17	0.410			
**SatO2/FiO2 (median, IQR)**	471	457–476	467	414–471	471	457–476	0.273			
**Respiratory support (any)**	214	24	9	36	205	24	0.158			
**Respiratory support type**							0.481			
*HFNC*	131	61	4	44	127	62				
*NIV*	23	11	1	11	22	11				
*O2 low flow*	60	28	4	44	56	27				
**BPEWS**							0.668			
*0–2*	716	94	18	100	698	93				
*3–4*	43	6	0		43	5				
*5*	2	<1	0		2	1				
≥6	4	<1	0		4	1				
**Device presence**										
*Any device*	281	32	15	60	266	31	**0.004**			
*Tracheostomy*	19	2	1	4	18	2	0.420			
*Gastrostomy/ileostomy*	84	9	3	12	81	9	0.723			
*Nasogastric tube*	67	8	5	20	62	7	**0.033**	**3.3 (1.2–8.9)**	**0.014**	**3.72 (1.30–10.57)**
*Central vascular line*	149	17	4	16	145	17	1.000			
*VP/VA/EVD shunt*	28	3	1	4	27	3	0.554			
*Thoracic/abdominal drainage*	14	2	1	4	13	2	0.330			
*Urinary catheter*	52	6	1	4	51	6	1.000			
*Peritoneal dialysis catheter*	3	<1	0		3	<1	1.000			
**Blood tests**										
*CRP, mg/dL (median IQR)*	0.8	0–4	0.9	0–10	0.8	0–4	0.462			
*PCT, ng/dL (median, IQR)*	0.2	0–1	0.4	0.1–2.6	0.2	0–0.9	0.190			
*WBC, el ×10^3^ (median, IQR)*	9.1	6–13	7.7	4.2–10	9.1	6–13	0.274			
*Neutrophils, el ×10^3^ (median, IQR)*	4.7	2.6–7.7	4.1	1.3–7.5	4.7	2.7–7.8	0.288			
*Lymphocytes, el ×10^3^ (median, IQR)*	2.2	1.2–4	1.6	1.2–2.3	2.2	1.2–4.1	0.074			
*Platelets, el ×10^3^ (median, IQR)*	290	202–403	245	210–330	290	200–410	0.160			
*Hb, g/L (median, IQR)*	117	103–130	112	970–130	120	103–130	0.573			
*Albumin, g/dL (median, IQR)*	3.7	3.2–4.1	3.5	3.1–3.8	3.7	3.2–4.1	0.109			
*Creatinine (median, IQR)*	0.3	0.2–0.5	0.3	0.2–0.4	0.3	0.2–0.5	0.379			
*Lactates (median, IQR)*	11	7–16	14	8–25	11	7–16	0.108			
**Outcome**										
*Deceased after 28 days*	11	1	5	20	6	7	**<0.001**			

**Table 2 jcm-14-06398-t002:** Generalized Linear Model for Factors Associated with PIMCU Length of Stay. PIMCU, pediatric intermediate care unit; PICU, pediatric intensive care unit; IQR, interquartile range; CI, confidence interval; IRR, incidence rate ratio; NT, not tested; HFNC, high-flow nasal cannula; NIV, non-invasive ventilation; VP, ventricular peritoneal; VA, ventricular-atrial drainage; EVD, external ventricular drainage; CRP, C-reactive protein; PCT, procalcitonin; WBC, white blood cells Hb, hemoglobin. Statistically significant values are expressed in bold.

	PIMCU Length of Stay (Days)		*p* Univariate	*p* Multivariable	IRR Multivariable (95% CI)
	Median	IQR			
**Total**	4	3–7			
**Gender**			0.922		
*Female*	4	3–7			
*Male*	4	3–7			
**Age (months) median IQR**	Spearman rho −0.074		0.173		
**Department of Origin**			**<0.001**		
*Lower intensity care units*	5	3–8			
*Emergency department*	4	3–6		**0.017** **(vs. Lower intensity)**	0.79 (0.64, 0.96)
*Operatory room*	8	8–8			
*PICU*	9	5.5–15.5			
**Reason for hospital admission**					
*Respiratory*	5	3–8	**0.008**	0.119	
*Cardiac*	4	3–6	0.839		
*Infectious*	4	2–6	**0.030**	0.422	
*Neurological/psychiatric*	4	3–5.5	**0.041**	0.470	
*Gastroenterological*	5	3–6	0.838		
*Surgical/orthopaedical*	4	3–9.5	0.506		
*Inflammatory disease*	5	4.5–6	0.583		
*Endocrine*	3	3–3	0.246		
*Cure prosecution*	7	5–12	0.140		
*Haemato-oncological*	5	4–7.5	0.272		
*Other*	5	3–6.5	0.827		
**Underlying disease**					
*Respiratory*	5	3–5.5	0.911		
*Cardiac*	4	3–7	0.899		
*Infectious*	6	5–6.5	0.682		
*Neurological/psychiatric*	4	3–6	0.533		
*Gastroenterological*	6.5	3–10	0.181		
*Surgical/orthopaedical*	7.5	4–14.5	**0.032**	0.470	
*Inflammatory disease*	6	4.5–6.5	0.705		
*Nephrological/Urological*	4	3.5–4.5	0.647		
*Endocrine*	3.5	3–4	0.259		
*Haemato-oncological*	5	4–7	0.090		
*Metabolic disease*	6.5	3–14.5	0.088		
*Prematurity*	7	4–12	**0.005**	0.205	
*Other*	5	3–7	0.640		
**Temperature at admission**			0.930		
<37 °C	4	3–7			
<38.5 °C	4	3–6			
>38.5 °C	5	3–7			
**Hypotension (adjusted for age)**	5	3–6	0.912		
**Tachycardia (adjusted for age)**	5	3–6	0.448		
**Bradycardia (adjusted for age)**	4	3–4	0.609		
**Tachypnea (adjusted for age)**	4	3–6.5	0.644		
**Bradypnea (adjusted for age)**	5	3–6	0.771		
**SatO2/FiO2 (median, IQR)**	Spearman rho −0.164		**0.002**	0.684	
**Respiratory support (any)**	6	4–9	**<0.001**	**0.035**	**1.36 (1.02, 1.81)**
**Respiratory support type**			0.161		
*HFNC*	7	5–9			
*NIV*	5	4–6			
*O2 low flow*	6	3.5–10			
**PEWS**			**0.010**		
*0–2*	4	3–7			
*3–4*	7	5–9			
*5*	6	6–6			
≥*6*	19	13–25			
**Device presence**					
*Any device*	7	4–11.5	**<0.001**		
*Tracheostomy*	3	2–3	0.053		
*Gastrostomy/ileostomy*	5	3–11	**0.017**	0.211	
*Nasogastric tube*	10	7.5–14	**<0.001**	**0.005**	1.63 (1.15, 2.3)
*Central vascular line*	8	5–14	**<0.001**	**<0.001**	1.73 (1.31, 2.27)
*VP/VA/EVD shunt*	6.5	5–7	0.296		
*Thoracic/abdominal drainage*	13	9–16	**0.002**	**<0.001**	3.17 (1.81, 5.56)
*Urinary catheter*	12.5	6–16	**<0.001**	0.608	
*Peritoneal dialysis catheter*			//		
**Blood tests**					
*CRP (median IQR)*	Spearman rho 0.192		**<0.001**	NT	
*PCT (median, IQR)*	Spearman rho 0.204		**0.009**	NT	
*WBC (median, IQR)*	Spearman rho 0.036		0.537		
*Neutrophils (median, IQR)*	Spearman rho 0.013		0.822		
*Lymphocytes (median, IQR)*	Spearman rho 0.102		0.079		
*Platelets (median, IQR)*	Spearman rho 0.118		**0.042**	NT	
*Hb (median, IQR)*	Spearman rho −0.136		**0.016**	NT	
*Albumin (median, IQR)*	Spearman rho −0.159		**0.027**	NT	
*Creatinine (median, IQR)*	Spearman rho −0.169		**0.005**	NT	
*Lactates (median, IQR)*	Spearman rho −0.110		0.086		

## Data Availability

The raw data supporting the conclusions of this article will be made available by the authors on request.

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
