# Peer review of "Optimizing Pediatric Intermediate Care: Clinical Predictors of Deterioration and Length of Stay in a Tertiary Setting"

_jcm, 2025, doi:10.3390/jcm14186398_

Round 1

Reviewer 1 Report

Comments and Suggestions for Authors

The study presents valuable research that resolves the essential clinical problem of finding predictors which lead to deteriorations and extended hospital stays in Pediatric Intermediate Care Unit (PIMCU). The study methodology fits well with the findings that the authors present in a structured manner. The research makes an essential contribution to this field since PIMCU-specific studies remain scarce. The following suggestions will help improve the quality of the manuscript.

Major revision

  1. Need deeper discussion on why BPEWS was not a predictor

    The research shows that Bedside Pediatric Early Warning Score (BPEWS) failed to predict patients who needed PICU transfer. These scoring systems exist to detect early clinical deterioration yet the results show they fail to achieve this purpose in this specific context. The discussion does not provide enough explanation regarding this important discovery.

Therefore, I suggest as follows: The discussion needs to include more details regarding the factors which may have caused BPEWS to fail as a predictive tool in this particular PIMCU group (e.g., PIMCU patients present with severe illness from the start, which lowers the predictive power of the score; site-specific patterns in BPEWS implementation may exist). This analysis would be highly advantageous for readers who wish to understand the constraints and proper implementation of early warning scores.

  1. Discussion of the statistical variable selection methodology

    The researchers restricted the multivariable model to variables which reached a p-value below 0.05 during univariable analysis. Research indicates that this approach sometimes misses essential confounding factors yet sometimes includes variables that reach statistical significance through random chance.

Therefore, I suggest as follows: Please explain the selection criteria for this variable selection approach in the Methods section or explain its possible drawbacks in the Discussion section. The explanation will improve readers' comprehension of statistical methods used in the study.

Minor revision

  1. Information on the excluded patient cohort

    The researchers eliminated 35 patients from planned admissions because their markedly higher baseline risk made them ineligible for analysis. The supplementary materials should include brief information about excluded patients' demographic features and main illnesses to enhance population understanding and minimize selection bias doubts.

  1. Clarification regarding the pre-existing conditions statement

    Discussion section reports that patients needing ICU admission had higher clinical complexity because pre-existing conditions occurred more frequently (p<0.001). Table 1 reveals no significant differences between the "early transfer (within 48 hours)" group and other groups regarding any pre-existing conditions. The explanation should specify whether the p<0.001 value refers to all 53 ICU transfers during the entire study period or to the comparison between patients with any pre-existing condition versus those without pre-existing conditions although this variable is not shown in Table 1.

Author Response

We thank reviewers for their constructive comments. Their insights have helped us improve the clarity, methodological transparency, and overall rigor of the work. We have addressed all points raised and revised the manuscript accordingly. Below, we provide point-by-point responses detailing the changes made and offering clarifications where appropriate.

  1. discussion on BPEWS:
    • We agree this is a critical finding. We have expanded our discussion to better explain why BPEWS might not be a predictor in our specific cohort.
  2. Discussion of the statistical variable selection methodology:
    • We acknowledge the limitations of our variable selection approach. We have added a statement in the "Materials and Methods" section to clarify. We have also added a sentence in the "Discussion" to explicitly state that this method may have excluded some important confounding factors, but our goal was to identify key associations within our specific dataset rather than building a universally applicable predictive model.
  3. Information on the excluded patient cohort:
    • As requested, we have added a brief description of the 35 excluded patients to the "Results" section.
  4. Clarification regarding the pre-existing conditions statement:
    • We apologize for the lack of clarity regarding the phrase "patients requiring ICU admission showed greater clinical complexity, with a higher prevalence of pre-existing conditions (p < 0.001) and medical devices at admission (p < 0.001)." This sentence was based on a previous analysis on total PICU transfers. We removed it to avoid confusion: we focus only on early PICU transfer: this is now clearly stated in the revised manuscript.

We believe that these revisions address all of your concerns and have substantially improved the quality of our manuscript. Thank you once again for your valuable insights.

Reviewer 2 Report

Comments and Suggestions for Authors

Your manuscript was a pleasure to read.  Well done!

I only have one question.  Please define what are included in"Surgical/orthopaedical".   Are orthopedic patients post surgery or not or both

Author Response

We thank reviewers for their constructive comments. Their insights have helped us improve the clarity, methodological transparency, and overall rigor of the work. We have addressed all points raised and revised the manuscript accordingly. Below, we provide point-by-point responses detailing the changes made and offering clarifications where appropriate.

  1. Definition of "Surgical/orthopaedical":
    • Thank you for your positive comments and for pointing this out. We have updated the "Materials and Methods" section to clarify "Surgical/orthopaedical" category.

Reviewer 3 Report

Comments and Suggestions for Authors

Dear Editor,

Thank you for the opportunity to review this manuscript, which seeks to identify admission factors associated with early unplanned transfer to the Pediatric Intensive Care Unit (PICU) within 48 hours, as well as factors associated with a prolonged length of stay (LOS) in the Pediatric Intermediate Care Unit (PIMCU). The study addresses clinically significant questions. However, several important issues need to be addressed

  1. Justification for the 48-Hour Cutoff:
    While early unplanned transfer to the PICU within 48 hours is undoubtedly important, the authors have not provided a rationale for choosing this specific cutoff over other possible timeframes. In this study, 53 patients required PICU admission, but only 25 were transferred within 48 hours. Please discuss the reasoning behind selecting the 48-hour cutoff and reference relevant guidelines from the Society of Critical Care Medicine to support this choice.

  2. Revision of Figure 1:
    Figure 1 should be redrawn as a detailed flowchart. It should clearly indicate where Univariate Analysis, Multivariable Analysis, and Generalized Linear Model were applied. This revision would help clarify the study design and enhance the transparency of the research methodology.

  3. Legend for Table 2 and Reporting of ROC Results:
    The legend for Table 2 is currently missing. In addition, to substantiate the credibility and clinical utility of the predictive models, it is important to report their performance. For the multivariable logistic regression model predicting early PICU transfer, the authors should include a measure of model discrimination, such as the Area Under the Receiver Operating Characteristic (AUC-ROC) curve. Including this information would significantly strengthen the manuscript.

  4. Discussion of an Important Limitation:
    A significant limitation that needs to be addressed is the small number of events for the primary outcome of early PICU transfer (n=25), resulting in a substantial class imbalance compared to the control group (n=868). This imbalance may affect the stability and generalizability of the regression model's findings. Furthermore, no propensity score matching (PSM) analysis was performed. The authors should explicitly acknowledge this limitation in the manuscript and discuss its potential impact on the results and their interpretation.

Author Response

We thank reviewers for their constructive comments. Their insights have helped us improve the clarity, methodological transparency, and overall rigor of the work. We have addressed all points raised and revised the manuscript accordingly. Below, we provide point-by-point responses detailing the changes made and offering clarifications where appropriate.

  1. Justification for the 48h Cutoff:
    • We appreciate this important question. We have added a rationale in the "Materials and Methods" section with reference.
  2. Revision of Figure 1:
    • We agree that a flowchart is a more effective way to present the study design. We have redrawn Figure 1 as a detailed flowchart that clearly illustrates the patient flow and the specific points where our statistical analyses were applied.
  3. Legend for Table 2 and Reporting of ROC Results:
    • We have added a clear legend to Table 2 as requested. We have also included the AUC-ROC curve in the "Results" section.
  4. Discussion of an Important Limitation:
    • We acknowledge the significant limitation of the class imbalance. We have added a paragraph to the "Limitations" section of the Discussion to explicitly address this issue and its potential impact on the stability and generalizability of our findings. We also note that, while we did not use propensity score matching, our primary goal was to identify associations within our unique retrospective dataset, not to create a broadly predictive model.

We believe that these revisions address all of your concerns and have substantially improved the quality of our manuscript. Thank you once again for your valuable insights.

Round 2

Reviewer 1 Report

Comments and Suggestions for Authors

Dear Authors, Your detailed response to my previous comments and your thoughtful modifications of the manuscript are greatly appreciated. I have examined your detailed responses together with the modified manuscript. Your responses have effectively resolved every point I raised. Your clarifications together with additional content have substantially enhanced both the quality and clarity of the paper. I appreciate your work on the BPEWS discussion section. The manuscript has become substantially stronger and I am confident it will provide significant value to the field. I appreciate your thorough efforts. I wish you the best of luck with the final decision.

Reviewer 3 Report

Comments and Suggestions for Authors

I have no more question at this stage